# *Mycobacterium marinum*: A Case-Based Narrative Review of Diagnosis and Management

**DOI:** 10.3390/microorganisms11071799

**Published:** 2023-07-13

**Authors:** Giorgos Tsiolakkis, Angelos Liontos, Sempastian Filippas-Ntekouan, Rafail Matzaras, Eleftheria Theodorou, Michail Vardas, Georgia Vairaktari, Anna Nikopoulou, Eirini Christaki

**Affiliations:** 1Department of Internal Medicine, Nicosia General Hospital, Nicosia 2029, Cyprus; tsiolakkisg7@gmail.com (G.T.); eriatheodorou@gmail.com (E.T.); 21st Division of Internal Medicine & Infectious Diseases Unit, University General Hospital of Ioannina, Faculty of Medicine, University of Ioannina, Str. Niarchou, 45500 Ioannina, Greece; angelosliontos@gmail.com (A.L.); sebastienfilippas@gmail.com (S.F.-N.); rafail.matz@gmail.com (R.M.); 3School of Medicine, University of Cyprus, Nicosia 2029, Cyprus; vardas.mic@gmail.com (M.V.); gina.vairaktari@gmail.com (G.V.); 4Department of Internal Medicine, G. Papanikolaou General Hospital of Thessaloniki, 57010 Thessaloniki, Greece; nikopanna@gmail.com

**Keywords:** mycobacteria, mycobacterial infections, *Mycobacterium marinum*, skin and soft tissue infection, skin lesions, non-tuberculous mycobacteria

## Abstract

Skin and soft tissue infections caused by non-tuberculous mycobacteria are occurring more frequently in recent years. However, chronic skin and soft tissue lesions present a challenge for clinicians, as the diagnostic work-up and definitive diagnosis require knowledge and available laboratory resources. We present here the case of a 66-year-old male patient who presented with painful abscess-like nodules on his right hand and forearm, which worsened after treatment with an anti-TNF-a agent. The fluid specimen taken from the lesion was positive for mycobacteria according to the acid-fast stain. *Mycobacterium marinum* was identified, first by next-generation sequencing and finally grown on culture, after eight weeks. Acknowledging the complexity of diagnosing and managing infections by non-tuberculous mycobacteria, and especially *Mycobacterium marinum*, we provide a review of the current epidemiology, clinical characteristics, diagnosis and management of *Mycobacterium marinum* infection.

## 1. Introduction

The investigation of chronic skin lesions is always challenging for a clinician, especially if an uncommon infection is suspected or the patient is immunocompromised. In that case, difficulty in reaching a diagnosis is not unusual because of the need for biologic specimen acquisition, the nature and the microbiologic characteristics of the implicating pathogens, and sometimes, the special laboratory equipment needed for an accurate diagnosis. *Mycobacterium marinum* (*M. marinum*) is a rare cause of chronic skin and soft tissue lesions. *M. marinum* is a non-tuberculous, slow-growing, acid-fast bacillus which causes a granulomatous tuberculosis-like illness in fish and other aquatic hosts. *M. marinum* is acquired from fish bites or contact with contaminated water. Most cases of human *M. marinum* infections are limited to the skin, although cases involving deep tissue and systemic illness have been described, especially in immunocompromised hosts. Diagnosis is difficult and requires a high index of suspicion, both from the clinician and from the lab. Treatment can be challenging and usually consists of a two or three drug combination.

We present the case of an *M. marinum* infection in a patient who presented with chronic skin and soft tissue lesions and who had received an anti-tumor necrosis factor alpha (anti-TNF-a) drug following a diagnosis of rheumatoid arthritis.

## 2. Case Presentation

A 66-year-old male patient was referred to the Infectious Diseases Outpatient Department for evaluation due to painful nodules on his right hand and forearm. Upon presentation, he had a painful ulcer on the second finger of the right hand, near the metacarpophalangeal joint. The ulcer appeared with raised borders, a dark-red color, and had a base crater with yellowish exudate. He also had several small to moderate size, painful, abscess-like lesions on his forearm, extending up to the elbow (Figure 1). Ipsilateral axillary lymphadenopathy (up to 2 cm) was noted. The rest of the physical examination was unremarkable. The patient reported no systemic symptoms, no fever, and denied pulmonary or gastrointestinal symptoms. He also denied recent travel and reported no animal contacts, no chemical or erosive agent exposures, and no sick contacts. The chest plain radiograph had no significant findings.

His past medical history was significant for spinal cord surgery (spondylodesis), hernia repair, left rib fractures, and gastroesophageal reflux disease. He denied smoking, and his current working activities were limited to office management in a private business. 

His present illness began about seven months ago when he was accidentally injured on the fourth finger of his right hand while he was cutting a diseased palm tree. He reported that the wound healed quickly but later developed a painful ipsilateral hand swelling. He received empirically a short course of antibiotics consisting of cefuroxime, clindamycin, and ciprofloxacin, without improvement. Two months after the initial injury, he consulted an orthopedic surgeon due to the persistent swelling of the right hand. The patient was diagnosed with carpal tunnel syndrome, for which he was operated, albeit with no improvement. Then, he consulted a rheumatologist, who diagnosed polyarthritis and prescribed corticosteroids, methotrexate, and leflunomide, again with no improvement. Of note, the patient denied any other local or systemic symptoms, and the laboratory exams were negative for antinuclear antibodies, rheumatoid factor, and anti-cyclic citrullinated peptide.

During the next two months, because of a lack of clinical improvement when on the disease-modifying antirheumatic drugs, the patient received three doses of infliximab. Approximately 10 days after this treatment, he noticed painful nodules on his right arm and forearm, extending up to the elbow. Some of the lesions automatically drained purulent material, and fluid specimens were sent for microbiologic examination. The Gram stain and culture were negative, but the acid-fast stain revealed many mycobacteria-like organisms (non-filamentous, non-branching). The polymerase chain reaction (PCR) for *Mycobacterium tuberculosis* (*M. tuberculosis*) was negative. Specific PCR for other mycobacteria was not available at our institution at that time.

There were no systemic symptoms and the laboratory work-up was only significant for C-reactive protein elevation (150 mg/dL, normal range < 5 mg/dL). A computed tomography scan of the right upper arm was performed to examine the extension of the lesions and revealed no involvement of deep structures or bone.

He was admitted to the hospital and was started on combination antimicrobials for non-tuberculous mycobacteria (NTM) causing skin and soft tissue infections. Initially, he was given imipenem/cilastatin intravenously in combination with clarithromycin orally, and later doxycycline and amikacin were added due to the slow response. While re-examining the patient’s history, he recalled that an artificial pool containing fish was located around the palm tree he was cutting. The suspicion of *M. marinum* infection was therefore high, although the culture remained negative. The antimicrobials were switched to ethambutol and clarithromycin.

The initial sample was sent to a reference laboratory for whole-genome sequencing analysis, which revealed the *Mycobacterium marinum*/*Mycobacterium ulcerans* (*M. ulcerans*) strains. The cultures became positive after eight weeks of incubation (solid medium incubating at 30 °C). The PCR on that specimen revealed the *M. ulcerans* strain; however, epidemiologically, that strain was the least possible, and because of the extensive genomic similarity between *M. marinum* and *M. ulcerans*, the specimen was sent again for a whole-genome sequencing test. The final analysis revealed *M. marinum*, which was sensitive to all the antimicrobials tested.

The clinical response was slow, and therefore, rifampicin 600 mg once daily orally was added to his treatment. Over the next few months, the patient had close follow-up, the lesions significantly improved until they disappeared and he had no recurrence. He received triple therapy for approximately three months and then clarithromycin as monotherapy for another two months, for a total duration of treatment of seven months.

## 3. Discussion and Review of the Literature

### 3.1. Microbiology and Epidemiology of M. marinum Infections

*M. marinum* is a non-tuberculous, non-motile, non-spore forming, photochromogen, slow-growing, acid-fast bacillus that is classified in group 1 of the Runyon’s classification system [1,2]. The pathogen grows best when incubated in a solid medium between 30 and 32 °C. It grows slowly, approximately between two and eight weeks. Unlike *M. tuberculosis*, *M. marinum’s* growth is inhibited at 37 °C [1,3].

*M. marinum* infection is uncommon. It was first described as a zoonotic disease in fish back in 1926. The first human infection was reported in 1951. International data concerning the incidence and prevalence of the infection are limited. The estimated annual incidence in the USA is 0.27 cases per 100,000 adult patients [1].

*M. marinum* is an opportunistic pathogen of fish and many other aquatic species, including mammals and amphibians, which serve as a reservoir, and it causes a granulomatous tuberculosis-like illness in these hosts [2]. *M. marinum* is acquired from fish bites or contact with contaminated water (salty or fresh), especially if the skin barrier is not intact. As a result, the major risk factors are occupational or environmental exposure, contact with fish tanks and aquariums, or injuries from tools and other equipment (for example, fishhooks) involved in these settings [4,5]. Many terms have been used in the literature to describe *M. marinum* infection. “Fish tank granuloma” and “aquarium granuloma” appear the most accurate, since exposure to these sources is a major risk factor for infection [3]. “Swimming pool granuloma” is somehow an obsolete term, since the widespread chlorination of swimming pools has diminished the exposure risk [3].

In the great majority of the human cases described in the literature, the pathogen causes skin infections, most commonly located in the upper limbs, especially on the hands and arms, spreading in a sporotrichosis-like pattern [4]. Infections occur in both immunocompetent and immunocompromised individuals, although the clinical course may be different. Immunosuppressive therapies increase the risk of mycobacterial infections, most commonly from *M. tuberculosis* but from NTM as well, and may reveal or exacerbate a subclinical underlying disease [6]. Some authors reported that the incidence of NTM is higher among such patients, especially in areas where the incidence of *M. tuberculosis* infection is low [7,8]. Multiple studies in the past years have established the specific correlation between anti-tumor necrosis factor alpha (anti-TNF-a) agents and mycobacterial infections [9]. This is because TNF-α is an important mediator in the complex procedure of granuloma formation that inhibits the dissemination of mycobacteria [10]. Most of these cases were reported in patients with rheumatoid arthritis, most probably because of the high incidence of the disease and the associated frequent use of TNF-α blocking agents for this indication [9,11].

### 3.2. Clinical Course and Complications

*M. marinum* infection in humans is primarily limited to the skin [1], comprising four clinical presentations [12]. The most common (~60% of cases) is the cutaneous form. In this form, infection presents as superficial cutaneous lesions, usually papulonodular, of the upper extremities, involving the hands and/or fingers, in 88.7–95% of cases [1,5,12,13]. In less than 1/4 cases, *M. marinum* spreads through the lymphatic vessels to the regional lymph nodes and advances in a sporotrichoid form [1,5,12,14]. Infiltration of the lymph nodes by *M. marinum* results in the development of multiple nodules (>3) mimicking sporotrichosis [1,5,12,14]. Less frequently, skin lesions may present as abscesses or plaques (granulomatous, pustular, ulcerative) [1,12]. Deeper infections caused by direct inoculation of the mycobacterium or spread from a cutaneous lesion occur in 20–40% of patients [1,12,15]. These include most commonly tenosynovitis, with or without cutaneous lesions, arthritis, bursitis, and osteomyelitis [1,12], and they could be associated with a delayed diagnosis. The disseminated form and systemic infection occur only in immunocompromised patients [1,2,12,13]. Solid-organ and hematopoietic stem cell transplantation and anti-TNF-a treatment increase the risk of systemic *M. marinum* infection [1,2,12,13]. Also, *M. marinum* infection has been reported in HIV patients during the pre-anti-retroviral therapy (ART) period. However, infection is uncommon in the post-ART era [16].

Most cases of the infection are associated with aquatic exposure (>80%). The hands and/or fingers are more frequently affected (89–95%) [1,13,15,17]. The risk of infection increases, owing to the higher exposure, when handling aquariums or fish. In addition, this risk increases as low temperatures favor bacterial growth [1,13,15,17]. Fish tank handling leads mostly to the cutaneous form of the infection. In contrast, marine activities (boating or fishing) are associated with deeper infections resulting from puncture wounds [12,15]. Thus, tenosynovitis in the hand or wrist is common due to the higher risk of penetrating injuries [18]. On the other hand, lesions of the lower extremities are commonly associated with swimming pools or indirect contact with other affected body areas [18].

Finally, less usual manifestations of *M. marinum* have been reported, such as orchitis/epididymitis, lesions mimicking lymphoma, and severe osteomyelitis [19,20,21,22]. Similarly, regional lymphadenopathy and deep organ infection, for example, of the lung and heart, are rare [23,24].

### 3.3. Diagnosis

*M. marinum* is suspected in patients with a compatible clinical picture and a history of injured skin that has been in contact with salty or fresh water.

The initial clinical and radiological findings are often non-specific [25]. The routine laboratory investigation is non-diagnostic, rheumatoid factor is negative and erythrocyte sedimentation rate (ESR) is usually normal or mildly elevated [26]. The tuberculin skin test using purified protein derivative is usually positive in 70–100% of the cases but is not specific [27,28]. In infections due to *M. marinum*, it is possible to have positive interferon gamma release assays due to cross-reactions between *M. marinum* and *M. tuberculosis* [1,29]. This is not confirmatory for the diagnosis, although it should raise the suspicion of *M. marinum* infection in patients presenting with skin infections who are negative for *M. tuberculosis*, especially in areas where the incidence of *M. tuberculosis* is low [4].

The mainstay of diagnosis is the isolation of *M. marinum* from fluid or tissue taken from skin or other lesions [1]. Optimally, specimens for diagnosis should be collected by means of biopsy from a nonulcerated area in close proximity to the lesion and not using superficial swabs, since microorganisms isolated with the latter method could represent colonization in most cases [1,14,15]. In some cases, the collection and examination of clinical specimens should be repeated to confirm or exclude the diagnosis of NTM infection [30].

Biopsy specimens should be sent for histopathologic examination, microscopy, culture, and molecular detection (PCR). Histologic examination is diagnostic for mycobacteriosis in 50% of cases since the findings depend on the age of the lesion [1]. The histopathologic examination can show noncaseating or necrotizing granulomas accompanied by neutrophil infiltration, in addition to lymphocytes, macrophages, and giant cells [25,31].

Microscopic examination after staining and culture using specific media are the cornerstones of mycobacterial identification. Microscopic examination alone is not able to distinguish between tuberculous and NTM species. In the vast majority of cases, the number of bacteria in clinical specimens is low, and thus, Ziehl–Neelsen staining of biopsy specimens or exudate from lesions is rarely positive. *M. marinum* is a catalase-positive, slow-growing, aerobic organism that grows optimally on Lowenstein–Jensen medium at 30 °C to 32 °C within 2 to 5 weeks and turns yellow with exposure to light [32]. Regarding its phenotypic and biochemical characteristics, *M. marinum* is photochromogenic (its colonies turn yellow–orange after light exposure), cannot produce nitrate reductase, and has the ability to grow on a medium containing thiacetazone [1]. Biosafety level 2 (BSL2) measures are required for the manipulation of NTM isolates [33]. Both liquid (Mycobacteria Growth Indicator Tube, MGIT) and/or solid media (Middlebrook 7H10 or 7H11 agar or Lowenstein–Jensen (L–J) agar) should be used for the culture of NTM. Ιn fact, ΝTM grow in a shorter period of time in liquid rather than in solid media [34]. Culture in liquid media is more sensitive and rapid than culture in solid media; however, the latter enables observation of the morphology and quantification of organisms [31]. If *M. marinum* infection is suspected, laboratory personnel should be informed so that the test can be performed under special temperature conditions [34]. Specimens should be incubated not only at 37 °C but also at 32 °C, as growth is limited at temperatures above 33 °C [31]. Cultures must be observed for 6–12 weeks [31,35]. A positive culture is the gold standard for the diagnosis of *M. marinum* as the cause of nodular skin lesions [1,25,36]. The positivity rate of the cultures is between 70 and 80%. Broth microdilution is the preferred method for determining susceptibility to antituberculosis drugs for a variety of NTM, including *M. marinum* [37].

The accurate and more rapid identification of mycobacterium species often requires molecular methods, according to the American Thoracic Society and Infectious Disease Society of America (ATS/IDSA). Molecular methods can be positive even in culture-negative cases. NTM with clinical significance should be identified to the species level [38]. Molecular methods entail 16S ribosomal DNA sequencing, high-performance liquid chromatography (HPLC), PCR-restriction length polymorphism analysis (PRA), and multi-gene and whole-genome sequencing, and they have resulted in a significant increase in NTM species identification from clinical specimens [38].

Examples of PCR-based methods coupled with reverse hybridization are the INNO-LiPA Mycobacteria v2 (Innogenetics), which is based on the amplification of the ribosomal gene spacer (16S–23S), and GenoType Mycobacterium CM/AS (Hain Lifescience), which is based on the amplification of the 23S rRNA gene [39,40]. However, these tests do not allow for differentiation between *M. marinum*, *M. ulcerans*, *M. ulcerans subsp*. *shinshuense*, *Mycobacterium shottsii*, and *Mycobacterium pseudoshottsii* [41]. Phylogenetically, *M. marinum* is very closely related to *M. ulcerans*, with an almost 99.6% identity based on their 16SrRNA gene sequence analysis [42,43,44]. Drawing a distinction between the two species is important because of the different clinical courses, treatment approaches and prognoses. The specific identification of *M. marinum* from *M. ulcerans* and from other mycobacterial species can be achieved by sequencing three hypervariable regions of the 16S rRNA gene [45].

More recently, matrix-assisted laser desorption ionization-time of flight mass spectrometry (MALDI-TOF MS) is increasingly being used for the identification of mycobacteria in many laboratories, with similar performance to conventional molecular methods [46]. Last, whole-genome sequencing (WGS), which allows the sequencing of the entire genome of multiple species in clinical samples, coupled with metagenomic approaches and available WGS-analysis tools, enables the direct identification and diagnosis of NTM and other pathogens [47].

To summarize, *Mycobacterium marinum* possess some features that may help distinguish *M. marinum* from other mycobacteria (tuberculous and NTM) as the cause of the infectious process. The first step in diagnosing *M. marinum* infection is to consider it as a probable cause of the patient’s symptoms, especially in view of a compatible history of aquatic exposure. Most clinical, laboratory and radiologic findings are non-specific. The tuberculin skin test is also non-specific, as it may be positive in all NTM, whereas IGRAs may have a positive predictive value in an appropriate clinical setting. Specimen microscopic examination alone is not able to distinguish between tuberculous and NTM species. Compared with other NTM, *M. marinum* grow optimally on Lowenstein–Jensen medium at 30 °C to 32 °C within 2 to 5 weeks and turn yellow with exposure to light, while other NTM grow optimally at 37 °C. Molecular methods can be used to accurately diagnose an NTM infection, even in culture-negative cases. As outlined above, special attention should be paid in order to correctly distinguish *M. marinum* from *M. ulcerans*, as these strains are closely related, with an almost 99.6% identity based on 16SrRNA gene sequence analysis.

### 3.4. Treatment and Prognosis

It has been shown that prompt evaluation of the clinical presentation exerts a prognostic role and could dictate the optimal treatment plan. *M. marinum* infection is rare, and as such, no comparative trials of skin and soft tissue infections have been carried out to date. The choice of an antimycobacterial regimen depends mostly on the choice and the experience of the treating physician rather than on hard evidence.

The cell wall of *M. marinum* is 10 times less permeable than the cell wall of *M. tuberculosis*, thus providing increased survival in unfavorable environments as well as decreased antibiotic penetration [1]. Indeed, regarding standard antimycobacterial drugs, *M. marinum* is intrinsically resistant to isoniazid and pyrazinamide [14,25,48]. On the other hand, *M. marinum* is susceptible to rifampin [49,50], rifabutin, and ethambutol [51], while it is moderately susceptible to streptomycin. Furthermore, *M. marinum* has been known to be susceptible to other antibiotic drug classes, such as macrolides (clarithromycin and azithromycin) [51], sulfonamides (trimethoprim/sulfamethoxazole) [52], doxycycline [50], minocycline [51], linezolid [48], and imipenem [48]. Cases treated with quinolones (such as levofloxacin and ciprofloxacin) have also been reported [53,54]. Considering the minimal inhibitory concentrations (MICs), rifampin and rifabutin are the most efficacious antibiotics. The MICs of clarithromycin, tetracyclines, linezolid, imipenem, cotrimoxazole, moxifloxacin, and amikacin are close to their breakpoints, and as such, these agents may be considered to have moderate activity [55,56,57]. What is hopeful, though, is that no description of acquired antibiotic resistance during treatment has been described so far for any of the aforementioned antibiotics. The minimal differences that have been observed in the MICs of the aforementioned antibiotics are usually due to misidentification or due to differences in the method of antibiotic susceptibility testing [58].

There are no published randomized controlled trials regarding the optimal treatment of *M. marinum* infection. According to the 2007 joint American Thoracic Society (ATS) and Infectious Diseases Society of America (IDSA) treatment guidelines [38], treatment should include at least two active agents for 1–2 months after symptom resolution. Typically, this consists of 3–4 months of treatment. Monotherapy may also be considered in patients with minimal disease or patients with superficial skin infection only. Susceptibility testing is not routinely recommended and should be reserved for patients who experience treatment failure on standard dual treatment.

It is not clear which is the optimal treatment for immunosuppressed patients, although interruption of immunosuppression along with dual or even triple treatment consisting of ethambutol plus rifampicin plus a third agent (minocycline, doxycycline, linezolid, clarithromycin) seems logical [45].

Surgical debridement may be considered as an adjunct to medical treatment in patients with extensive disease, in patients with deep structure disease (such as osteomyelitis, tenosynovitis, arthritis, and in closed spaces of the hand), and in patients who fail to respond to standard treatment [59,60]. Adjunctive photodynamic therapy has been used in some antibiotic-resistant cases with promising results [61,62].

In a study of 63 patients across French hospitals [5], the median duration of antibiotics was 3.5 months, with a significantly longer treatment duration in patients with deep structure disease compared with patients with skin disease. Monotherapy was given to 23 patients (37%), while combination treatment was given to 40 patients (63%). The most frequent drug combination was clarithromycin plus rifampin (n = 20). Tetracyclines (doxycycline and minocycline) were mainly prescribed to patients with skin and soft tissue disease. Treatment failure was observed in eight (13%) patients. Failure was significantly related to deep structure disease (*p* = 0.04) and to the skin lesion aspect of the ulcer (*p* = 0.02). Failure was not related to any specific treatment regimen or to the duration of treatment.

In a retrospective study of 53 patients with *M. marinum* infections in Denmark [13], monotherapy was given in 45.3% of cases. The most common antibiotic given as monotherapy was doxycycline (36.4%), while the median duration of treatment was 91 days. Considering combination treatment, rifampicin plus clarithromycin was the most frequently used combination treatment (38.9%). The median treatment duration for patients with deep tissue infection was 240 days, while all the patients received a combination treatment of at least clarithromycin and ethambutol. Only six patients required surgical treatment. In a recent review [1], it was recommended that treatment be administered for at least 2 months after the healing of the mycobacterial lesions.

In another retrospective study of 18 patients [63], monotherapy was administered to 14 patients (11 received clarithromycin, while the remaining 3 received initially doxycycline), while combination treatment was administered to the remaining 4 patients. All the evaluable patients (5/18 were lost to follow up) were successfully treated.

On the other hand, in another study [15], a single antibiotic regimen was used in 5/17 (29%) patients. Combination treatment with 2 drugs was used in 7/17 (41%) patients and a triple-drug regimen was used in 5/17 (29%) patients. Clarithromycin, azithromycin, ethambutol, and rifampicin were the most frequently used antibiotics, either as monotherapy or as a combination treatment. Moxifloxacin and trimethoprim/sulfamethoxazole were also used in a minority of patients. The median treatment duration was 5 months, irrespective of the extent of the disease. Unexpectedly, surgical intervention was required in 22/28 patients (79%), which was more common than in the other case series reported so far. Summary of the studies is presented in Table 1. 

In summary, no definite conclusions regarding the optimal treatment regimen can be drawn until controlled trials have been carried out, which is difficult due to the rarity of the disease. It is reasonable to follow the 2007 ATS/IDSA guidelines regarding a treatment duration of 1–2 months after symptom resolution. Treatment with a single-drug regimen can be administered in mild cases which involve only superficial structures. Either tetracycline or clarithromycin seems a reasonable choice. Regarding combination treatment, clarithromycin plus ethambutol seems to be a reasonable option. Rifampin may be combined with clarithromycin or added to the above regimen in patients with deep structure disease due to its better penetration. Surgical evaluation and treatment should be offered to patients with deep structure and extensive disease as an adjunct to antibiotic treatment.

Most small ulcers or nodules are self-healed without treatment in 1 to 2 years [60,64]. The prognosis of treated *M. marinum* infection in immunocompetent patients is excellent [16]. Treatment failure is uncommon (5.7–13%), and it is mostly due to deeper tissue infection and ulcerative skin lesions rather than to antimicrobial resistance, which is low [16]. Skin-localized infection often resolves after 1–2.5 months of antibiotic treatment. In contrast, treatment of deeper tissue infection takes significantly longer (up to 18 months) [1]. Deeper infections typically require surgical procedures compared with the cutaneous form [60,65], especially in immunocompromised patients [27]. This should aim to limit the spread of the infection and preserve the function of the hand, whereas additional procedures may increase the risk of postoperative morbidity and scarring [66,67]. It is evident that as the time to diagnosis increases, the burden of disease is greater, and this can result in repeated office visits, prolonged antibiotic treatment, repeated surgery with concomitant immobilization, and the need for hospitalization [5,12].

**Table 1 microorganisms-11-01799-t001:** Summary of the studies.

Study	N	Treatment Regimen	Duration	Response
[5]	63	Monotherapy in 37% (minocycline/doxycycline or clarithromycin) Combination in 63% (the most frequent was clarithromycin + rifampin)	3.5 months	87% were cured, 13% experienced treatment failure
[13]	53	Monotherapy in 45.3% (the most usual drug was doxycycline) Combination in 54.7% (the most usual combination was rifampin + clarithromycin)	87 days (median) for superficial infection 240 days for deep tissue infection	75.5% were improved, 5.7% were classified as treatment failure
[63]	18	14/18 received monotherapy (11 received clarithromycin, 3 received doxycycline/minocycline) 4/18 received combination (clarithromycin, ethambutol and rifampin)	10 weeks (median)	13/18 were classified as successful, 5/18 were lost to follow up
[15]	28	5/28 received monotherapy 12/28 received combination treatment (the most common were ethambutol, rifampin, clarithromycin, azithromycin, and moxifloxacin) 11/28 received unknown antibiotics	5 months (median)	21/28 were improved, 7/28 were lost to follow up

## 4. Conclusions

In this case-based review, we have highlighted, on many levels, the challenges of managing skin and soft tissue *M. marinum* infections. First, it has depicted the importance of taking a detailed history, as this will raise the clinical suspicion of *M. marinum* and will correctly guide the diagnostic work-up. More specifically, the association of anti-TNF-a blockade and NTM infections is increasingly recognized. Second, it has illustrated how demanding the laboratory confirmation of *M. marinum* can be, as classic methods may need to be coupled with novel molecular or sequencing techniques. Last, despite the slow response to treatment, a combination of appropriate antibiotics are effective and most frequently result in the complete resolution of the (even potentially extensive) cutaneous manifestations.

## Figures and Tables

**Figure 1 microorganisms-11-01799-f001:**
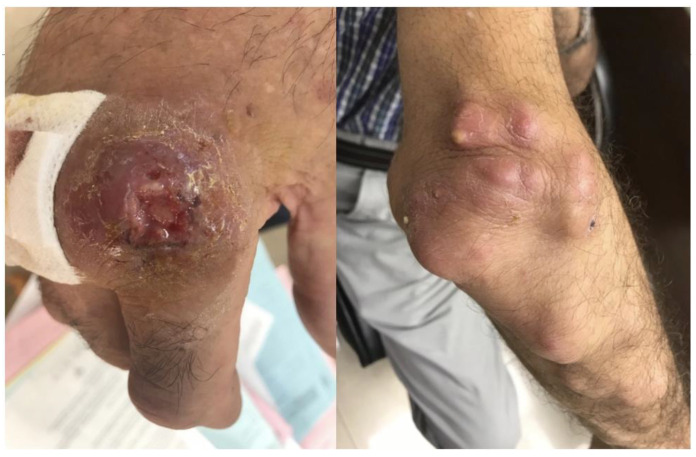
Skin ulcer on the second finger of the right hand, near the metacarpophalangeal joint, with raised borders, a dark-red color, and a base crater with yellowish exudate. Also illustrated are several small to moderate size abscesses on the right forearm, extending up to the elbow, with lymphangitic spread.

## Data Availability

No new data were created or analyzed in this study. Data sharing is not applicable to this article.

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
