# Peer review of "Mycobacterium marinum: A Case-Based Narrative Review of Diagnosis and Management"

_microorganisms, 2023, doi:10.3390/microorganisms11071799_

Round 1

Reviewer 1 Report

The manuscript entitled “Mycobacterium marinum: A narrative review of diagnosis and management, based on a case.” provides a comprehensive summary of the epidemiology, clinical features, diagnosis, and management of Mycobacterium marinum infection, offering valuable insights for diagnosing this condition. However, there are several areas in the manuscript that require improvement. 

1. The abstract section contains excessive details about the case, some of which do not directly contribute to the main content of the manuscript. The abstract should focus on highlighting the topic and the motivation of the review. The authors should revise and refine the abstract to better reflect the topic of the manuscript.

2. The introduction section provides limited and one-sided information. It should include a brief overview of the basic characteristics of Mycobacterium marinum, highlighting the main topic of the manuscript. The authors should revise this section to provide a more comprehensive and balanced overview of Mycobacterium marinum, focusing on the main topic of the manuscript.

3. The presentation of the case lacks crucial results, such as sequencing, PCR, and acid-fast staining results. It is essential to supplement these key results to support the conclusion that the observed case was caused by Mycobacterium marinum infection. Relying solely on gross lesion images is insufficient. The authors should include these important results to strengthen their conclusion regarding the Mycobacterium marinum infection in the presented case.

4. Section 3.2 mentioned prognosis, while section 3.4 also mentioned prognosis. The authors should consolidate all information related to prognosis into section 3.4.

5. The review lacks a section specifically addressing the distinguishing diagnostic characteristics of Mycobacterium marinum infection compared to other mycobacterial infections. The authors should revise the manuscript to incorporate a dedicated section that highlights the unique diagnostic characteristics of Mycobacterium marinum infection in comparison to other mycobacterial infections.

These are the major issues in this manuscript that need to be appropriately addressed before considering for publication.

Author Response

Reviewer 1
The manuscript entitled “Mycobacterium marinum: A narrative review of diagnosis and management, based on a case.” provides a comprehensive summary of the epidemiology, clinical features, diagnosis, and management of Mycobacterium marinum infection, offering valuable insights for diagnosing this condition. However, there are several areas in the manuscript that require improvement. 
1.    The abstract section contains excessive details about the case, some of which do not directly contribute to the main content of the manuscript. The abstract should focus on highlighting the topic and the motivation of the review. The authors should revise and refine the abstract to better reflect the topic of the manuscript.

Answer to Comment 1: We would like to thank the reviewer for his/her valuable comment. We have revised the abstract to make the content more concise and relevant to the aim of the review (lines 16-27).

2. The introduction section provides limited and one-sided information. It should include a brief overview of the basic characteristics of Mycobacterium marinum, highlighting the main topic of the manuscript. The authors should revise this section to provide a more comprehensive and balanced overview of Mycobacterium marinum, focusing on the main topic of the manuscript.

Answer to the Comment 2: We would like to thank the reviewer for his/her thoughtful comment. We have revised the introduction section and we have included a brief overview of Mycobacterium marinum characteristics to ameliorate this section of the manuscript (lines 37-44). 

2.    The presentation of the case lacks crucial results, such as sequencing, PCR, and acid-fast staining results. It is essential to supplement these key results to support the conclusion that the observed case was caused by Mycobacterium marinum infection. Relying solely on gross lesion images is insufficient. The authors should include these important results to strengthen their conclusion regarding the Mycobacterium marinum infection in the presented case.

Answer to the Comment 3: We would like to thank the reviewer for asking us to clarify the steps we took for appropriate diagnosis of Mycobacterium marinum in our patient. The information about acid-fast, PCR and sequencing data are presented in section 2 (lines 78-82 and 96-103). 

4. Section 3.2 mentioned prognosis, while section 3.4 also mentioned prognosis. The authors should consolidate all information related to prognosis into section 3.4.

Answer to the Comment 4: We would like to thank the reviewer for the comment. The title information has been updated as suggested by the reviewer. Text in sections 3.2 and 3.4 were reviewed for information regarding prognosis and this was merged in section 3.4.

5. The review lacks a section specifically addressing the distinguishing diagnostic characteristics of Mycobacterium marinum infection compared to other mycobacterial infections. The authors should revise the manuscript to incorporate a dedicated section that highlights the unique diagnostic characteristics of Mycobacterium marinum infection in comparison to other mycobacterial infections.

Answer to the Comment 5: We would like to thank the reviewer for this important point. We have added a section that summarizes the distinguishing diagnostic characteristics of Mycobacterium marinum infection compared with other mycobacterial infections at the end of section 3.3, as recommended by the reviewer (lines 257-270).

Reviewer 2 Report

Dear Editor, Dear authors, 
Thank you for inviting me to review this manuscript, a case report of a M. marinum cutaneous infection with a literature review. The manuscript is solidly organized and clear in its presentation and intents, I have some minor comments. 
1) Why was rifampicin added to the regimen only later on since, as outlined in the discussion it is the most active agent? Please clarify. 
2) It would be useful to have a table summirising the studies addressing treatment mentioned in the discussion comprising treatment regimens, duration, response. 

Author Response

Reviewer 2
Thank you for inviting me to review this manuscript, a case report of a M. marinum cutaneous infection with a literature review. The manuscript is solidly organized and clear in its presentation and intents, I have some minor comments.

1) Why was rifampicin added to the regimen only later on since, as outlined in the discussion
it is the most active agent? Please clarify. 
2) It would be useful to have a table summirising the studies addressing treatment mentioned in the discussion comprising treatment regimens, duration, response. 

Answer to Comment 1: Since there are no solid guidelines for M. Marinum (first and second line) treatment recommendations, and based on existing evidence, also taking into account the patient’s comorbidities and laboratory tests, we decided to initiate clarithromycin and ethambutol. However, an alternative treatment combination could have included rifampin from the start and it would have been equally or more justified.

Answer to Comment 2: We would like to thank the reviewer for highlighting this important point. We have added a table comprising of the treatment regimens, duration and treatment response of the studies referenced in the manuscript, as suggested by the reviewer.

Round 2

Reviewer 1 Report

NO